# Diverging roles of TRPV1 and TRPM2 in warm-temperature detection

**Muad Y Abd El Hay[1,2]\*, Gretel B Kamm[1], Alejandro Tlaie Boria[2], Jan Siemens[1,3]\***

[1]Department of Pharmacology, Heidelberg University, Heidelberg, Germany; [2]Ernst Strüngmann Institute for Neuroscience in Cooperation with the Max Planck Society, Frankfurt am Main, Germany; [3]Molecular Medicine Partnership Unit, European Molecular Biology Laboratory (EMBL), Heidelberg, Germany

## eLife Assessment

In this article, Abd El Hay and colleagues use an innovative behavioural assay and analysis method, together with standard calcium imaging experiments on cultured dorsal root ganglion (DRG) neurons, to evaluate the consequences of global knockout of TRPV1 and TRPM2, and overexpression of TRPV1, on warmth detection. **Compelling** evidence is provided for a role of TRPM2 channels in warmth avoidance behaviour, but it remains unclear whether this involves channel activity in the periphery or in the brain. In contrast, TRPV1 is clearly implicated at the cellular level in warmth detection. These findings are **important** because there is substantial ongoing discussion regarding the contribution of TRP channels to different aspects of thermo-sensation.

**\*For correspondence:**
muad.abdelhay@gmail.com (MYAEH);
jan.siemens@pharma.uni-heidelberg.de (JS)

**Competing interest:** The authors declare that no competing interests exist.

## Abstract
The perception of innocuous temperatures is crucial for thermoregulation. The TRP ion channels TRPV1 and TRPM2 have been implicated in warmth detection, yet their precise roles remain unclear. A key challenge is the low prevalence of warmth-sensitive sensory neurons, comprising fewer than 10% of rodent dorsal root ganglion (DRG) neurons. Using calcium imaging of >20,000 cultured mouse DRG neurons, we uncovered distinct contributions of TRPV1 and TRPM2 to warmth sensitivity. TRPV1's absence – and to a lesser extent absence of TRPM2 – reduces the number of neurons responding to warmth. Additionally, TRPV1 mediates the rapid, dynamic response to a warmth challenge. Behavioural tracking in a whole-body thermal preference assay revealed that these cellular differences shape nuanced thermal behaviours. Drift diffusion modelling of decision-making in mice exposed to varying temperatures showed that TRPV1 deletion impairs evidence accumulation, reducing the precision of thermal choice, while TRPM2 deletion increases overall preference for warmer environments that wildtype mice avoid. It remains unclear whether TRPM2 in DRG sensory neurons or elsewhere mediates thermal preference. Our findings suggest that different aspects of thermal information, such as stimulation speed and temperature magnitude, are encoded by distinct TRP channel mechanisms.

## Introduction

The detection of temperature and the related behavioural responses are an integral part of our sensory interaction with the outside world. Thus, it is not surprising that temperature detection was one of the first sensory modalities to be studied in contemporary neuroscience (*Zotterman, 1936*). Early studies concentrated on the characterization of temperature-specific sensory fibres, covering the range from noxious cold, through innocuous cold and warm, to noxious heat (*Hensel and Schafer, 1984*). However, the molecular mechanism by which temperature activates these fibres remained elusive for decades.

A major breakthrough in the field of somatosensory research was the identification of temperature-sensitive ion channels that belong to the transient-receptor potential (TRP) super family as the molecular sensors responsible for the detection of noxious cold and heat in sensory neurons (*Caterina et al., 1997*; *Peier et al., 2002*; *McKemy et al., 2002*). However, the detection of temperatures in-between noxious cold and heat (25– 43°C), which are often perceived as non-painful, is incompletely understood. This innocuous temperature range also contains the so-called thermoneutral point (TNP), an ambient temperature (29–33°C) at which mice do not exert additional energy to maintain their body temperature (*Škop et al., 2020*). This makes the innocuous temperature range crucial for thermoregulation and the animal's subsequent thermal or comfort choice. Recent studies began to uncover the mechanisms behind innocuous warm-temperature detection, thereby mainly converging on three candidate cation channels, namely TRPV1, TRPM2, and TRPM8 (*Yarmolinsky et al., 2016*; *Tan and McNaughton, 2016*; *Vilar et al., 2020*; *Mulier et al., 2020*; *Paricio-Montesinos et al., 2020*). Interestingly, the evidence for the involvement of TRPV1 and TRPM2 channels is seemingly contradictory.

TRPV1 is traditionally associated with the response to noxious temperature stimuli (>42°C), with the ability to become sensitive to lower temperatures in inflammatory contexts (*Caterina et al., 1997*; *Tominaga et al., 1998*; *Caterina et al., 2000*; *Davis et al., 2000*). However, in vivo calcium imaging of trigeminal sensory neurons in animals lacking TRPV1 showed a complete absence of responses to warm stimuli applied to the oral cavity of mice, while responses to hot temperatures were unchanged. Furthermore, acute inhibition of TRPV1 in animals trained to discriminate innocuous cold from warmth through a nose port led to a reduction in their performance (*Yarmolinsky et al., 2016*). A similar result was described in another operant behaviour task where animals had to report a warming stimulus applied to the paws (*Paricio-Montesinos et al., 2020*). These results stand in contrast to other studies that showed no involvement of TRPV1 in thermal preference across the warm-temperature range (*Shimizu et al., 2005*; *Pogorzala et al., 2013*; *Marics et al., 2014*).

TRPM2 has reported ex vivo activation temperatures between 35°C and >40°C, depending on the cellular context, and was first described as a physiological temperature sensor in pancreatic islet cells (*Togashi et al., 2006*; *Bartók and Csanády, 2022*). Calcium imaging of dorsal root ganglion (DRG) cultures from animals lacking TRPM2 showed a reduction in the proportion of warm- and heat-responsive neurons in comparison to wildtype sensory neuron cultures (*Tan and McNaughton, 2016*). Interestingly, *Trpm2$^{-/-}$* animals are unable to differentiate temperatures across the innocuous warm range in thermal preference tasks (*Tan and McNaughton, 2016*; *Ujisawa et al., 2022*), while their ability to avoid noxious temperatures is not affected. Interestingly, and similar to TRPV1, *Trpm2*-deficient animals trained to report warming of their paws were less sensitive than wildtype animals (*Paricio-Montesinos et al., 2020*).

In summary, both for TRPV1 and TRPM2, there is an apparent disconnect between the observations in cellular assays and the behavioural tasks assessing temperature detection. In vivo calcium imaging coupled with warm-temperature stimuli only shows a relevance of TRPV1, but not TRPM2 in the innocuous temperature range (*Yarmolinsky et al., 2016*). Contrary to that, the lack of TRPM2, but not TRPV1, more consistently affects warm-temperature detection in assays of temperature preference (*Tan and McNaughton, 2016*; *Pogorzala et al., 2013*; *Shimizu et al., 2005*; *Marics et al., 2014*), with both affecting the animals' temperature perception in operant behavioural assays, albeit only subtly (*Yarmolinsky et al., 2016*; *Paricio-Montesinos et al., 2020*).

One main challenge for the analysis of warm-sensitive neurons, neturons that respond to innocuous temperature stimuli between 25°C and 43°C, is the low abundance of this neuronal population, which represents about 3–10% of sensory neurons in rodents (*Wang et al., 2018*; *Tan and McNaughton, 2016*). Using in-depth functional analysis of thousands of sensory neurons from multiple animals, we here describe that TRPV1 and TRPM2 are both involved in the detection of innocuous (warm) temperature stimuli. Furthermore, we demonstrate the diverging roles both channels play in warm-temperature detection through a novel thermal preference behaviour assay.

# Results

## The thermal chamber preference test allows precise discrimination of subtle temperature differences in the innocuous range

### Mice prefer 31°C to warmer temperatures when ambient and floor temperature are controlled

The ability to avoid uncomfortable environmental temperatures or move towards pleasant thermal conditions is fundamental to sustaining life. This process is known as behavioural thermoregulation and can be found in most groups of animals (*Mota-Rojas et al., 2021*). The preference of rodents for temperature is traditionally assessed via paradigms that challenge the animals with differing floor temperatures (*Gordon et al., 1998*; *Moqrich et al., 2005*; *Touska et al., 2016*). This leads to preference development that is based on temperature detection through glabrous skin, such as the paws, tail, and nose. Animals, however, are capable of integrating temperature from glabrous and non-glabrous skin (*Romanovsky, 2014*). With the aim to probe behavioural thermoregulation in a more holistic context, we developed a thermal discrimination assay where both the floor and ambient temperature are controlled, termed the thermal chamber preference test (CPT) (*Figure 1A–D*, *Figure 1—figure supplement 1*). When presented with warm ambient temperatures (34°C and 38°C) and 31°C as control temperature, mice significantly preferred the 31°C chamber over the warmer chambers (*Figure 1E and F*). Compared to the classic two-plate preference test (TPT), wildtype animals developed a stronger preference for the 31°C side in the CPT (*Figure 1—figure supplement 1I*). Additionally, animals showed a clear preference for 31°C when given 34°C as an option in the CPT. This is not observed in the classic TPT (*Figure 1—figure supplement 1I*). This observation suggests that more subtle ambient temperature differences, relating to comfort and thermoregulation, are more faithfully assessed in the CPT assay.

### TRPM2 is necessary for establishing a preference in the warm-temperature range

Previous studies assessing warm-temperature detection using the TPT with animals lacking TRPV1 or TRPM2 showed that *Trpm2*[-/-] animals failed to differentiate between 31°C and 38°C (reproduced in *Figure 1—figure supplement 1*) while *Trpv1*[-/-] animals, similar to wildtypes, preferred the 31°C side (*Tan and McNaughton, 2016*; *Pogorzala et al., 2013*). Similarly, when using the newly developed CPT, *Trpv1*[-/-] animals showed a similar temperature preference to wildtypes while *Trpm2*[-/-] animals failed to develop a preference for the thermoneutral (31°C) side (*Figure 1E and F*), without affecting their preference at 25°C (*Figure 1—figure supplement 2A and C*). In addition to the previously described phenotype at 38°C, *Trpm2*[-/-] animals were also unable to discriminate 34°C from 31°C, emphasizing the relevance of TRPM2 at milder warm temperatures (*Figure 1E and F*, *Figure 1—figure supplement 1*). Notably, the phenotype of *Trpm2*[-/-] animals was similar to that of animals lacking most, if not all, peripheral (heat and cold) thermosensors (*Trpv1-Abl*, *Mishra et al., 2011*, *Figure 1—figure supplement 1J and K*). These results confirm previous data demonstrating the requirement for TRPM2 rather than TRPV1, in preference development in the warm-temperature range (*Tan and McNaughton, 2016*; *Pogorzala et al., 2013*).

### TRPV1 and TRPM2 affect different aspects of warm-temperature detection

Traditionally, analyses of temperature preference are limited to reporting the proportion of time an animal spent at the test temperature, without assessing more fine-grained thermal preference behaviour, such as the sequence of chamber crossings and intermittent pauses (visit lengths, *Moqrich et al., 2005*; *Touska et al., 2016*; *Gordon et al., 1998*). We observed that in the CPT, animals cross from one chamber to the other, probing the chamber, before crossing back (*Figure 1G*). We quantified the number of crossings and the lengths of these episodes throughout the experiments (*Figure 1H*). At the start of the experiment, mice of all genotypes crossed more often than at the end of the experiment while maintaining similar durations of their visits to the warmer chamber (*Figure 1H–J*). *Trpv1*[-/-] animals showed a significantly higher crossing rate compared to wildtype animals at both 34°C and 38°C (*Figure 1H and J*). *Trpm2*[-/-] animals, on the other hand, had significantly longer visits to the warmer chamber compared to wildtype animals, while having either similar (at 38°C) or a reduced crossing rate (at 34°C) (*Figure 1H–J*).

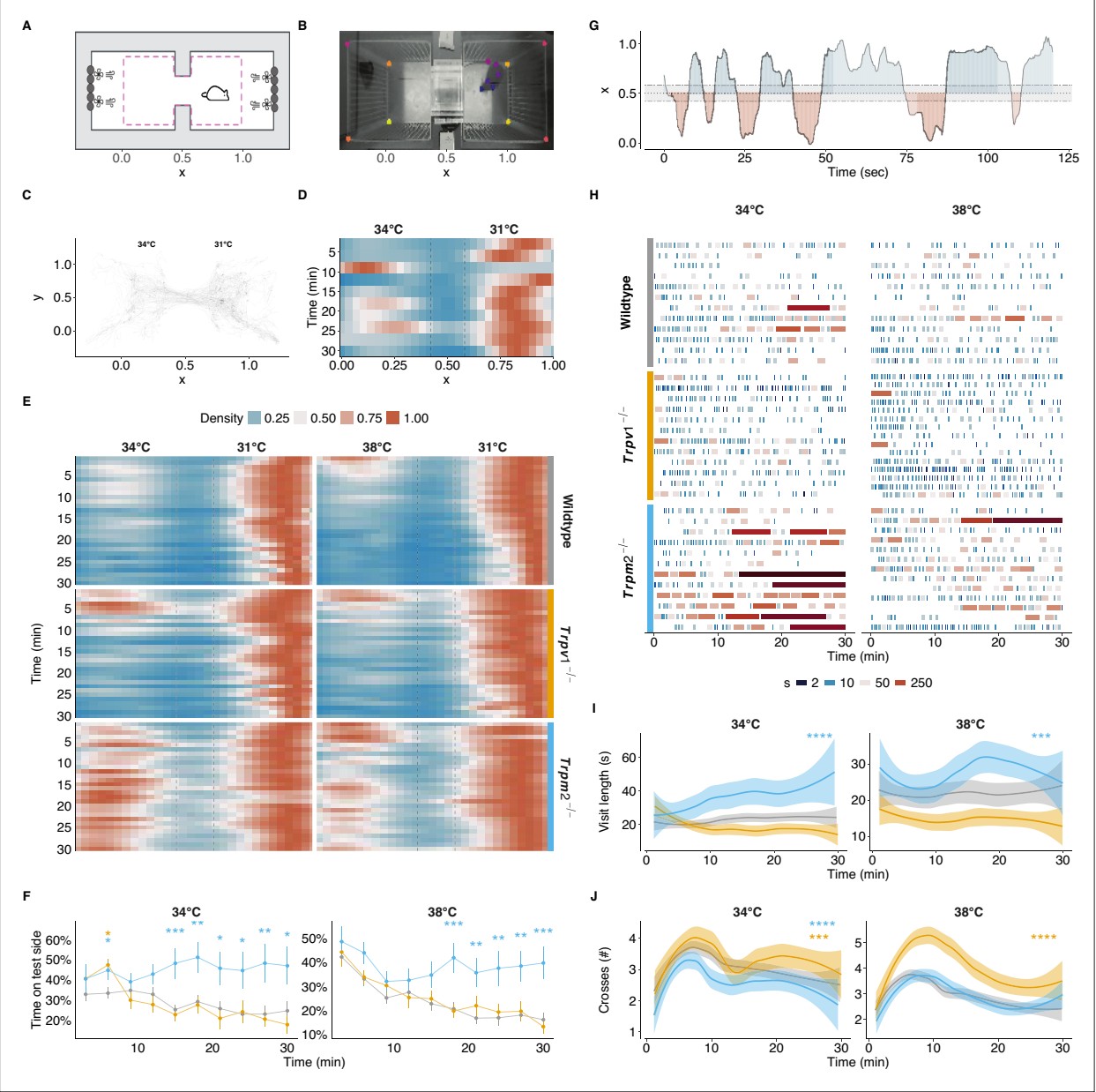

**Figure 1.** A novel ambient temperature preference test. (**A**) Schematic of the chamber preference test from the top. Grey outlines the outer enclosure and the dashed line the internal cage. Peltier elements (grey oval shapes) were combined with fans for precise control of the temperature. See *Figure 1—figure supplement 1* for a more detailed view. (**B**) A representative image of an animal exploring the chambers. Coloured dots represent the tracked keypoints on the animal and reference points in the enclosure. (**C**) Tracking of an example animal for 30 minutes at 31°C (right chamber) and 34°C (left chamber). (**D**) Density maps of the x-position of the animal in (**C**) over 30 minutes; binned in 3-minute-long intervals. Dashed lines represent the tunnel connecting both chambers. (**E**) Density maps as in (**D**) with 1-minute bins of all animals from wildtype (n = 48), *Trpv1⁻/⁻* (n = 15), and *Trpm2⁻/⁻* (n = 28) genotypes. (**F**) Proportion of time spent in the test chamber for animals shown in (**E**) over time, binned in 3-minute-long intervals. Mean and standard error of the mean (SEM) shown. ANOVA over genotype (34°C: $F_{(2,70)}$ = 7.30, p=0.001; 38°C: $F_{(2,76)}$ = 7.84, p<0.001), time (34°C: $F_{(5.63,394.35)}$ = 2.51, p<0.05; 38°C: $F_{(3.86,293.72)}$ = 11.07, p<0.001), and their interaction (34°C: $F_{(11.27,394.35)}$ = 1.98, p<0.05). Results of post hoc multiple comparison by timepoint against wildtype are indicated. (**G**) Exemplary behaviour of the animal in (**C**) and (**D**) over the first 120 seconds of the experiment, highlighting the visit frequency and duration of time spent in each chamber. The dashed line represents the tunnel connecting the chambers. (**H**) Overview of the frequency and length of the visits to the test chamber for 15 randomly sampled animals per genotype, shown in (**E**). Each visit is coloured by the log2 of its length to highlight varying visit lengths. (**I**) Averaged and smoothed visit lengths in a 3-minute rolling window with a 1-minute lag. The shaded area represents the 95% confidence interval. Linear mixed model over genotype and time with random effects across animals. (**J**) Averaged and smoothed number of crosses in a 3-minute rolling window. The shaded area represents the 95% confidence interval. Cox regression

*Figure 1 continued on next page*

*Figure 1 continued*

over genotype (34°C $X^2$ (2) = 49.67, p<0.001; 38°C $X^2$ (2) = 55.74, p<0.0001). **(I, J)** Results of post hoc multiple comparison against wildtype are indicated. *p<0.05, **p<0.01, ***p<0.001, ****p<0.0001.

The online version of this article includes the following figure supplement(s) for figure 1:

**Figure supplement 1.** Establishing a novel ambient temperature preference test.

**Figure supplement 2.** Temperature preference behaviour at 25°C.

**Figure supplement 3.** Temperature preference behaviour of *Trpm8-/-* animals.

Notably, animals lacking TRPM8, a cold-sensitive TRP-channel that was shown to be critical for learning to report warming in an operant behavioural assay (*Paricio-Montesinos et al., 2020*), showed similar preference behaviour to wildtype animals across the warm-temperature range, but increased visit lengths at 25°C and fewer crosses across all tested temperatures (*Figure 1—figure supplement 3A–E*).

Together, these observations suggest that, contrary to previous findings using the TPT, both TRPV1 and TRPM2 contribute to the animals' ability to detect warm temperatures and to drive associated thermal preference behaviours, albeit the two ion channels modulate different aspects of the behavior.

## *Trpv1* knockout mice have decreased proportions of warm-sensitive neurons

### A small subpopulation of cultured primary sensory neurons responds to warm temperatures

*Trpv1* is highly expressed in peripheral sensory neurons that reside bilaterally in so-called DRG along the spinal cord. To assess the individual contribution of *Trpv1* and *Trpm2* channels to ambient warm-temperature detection, and to account for the integration of temperature across the whole body of the animal, we cultured primary DRG neurons pooled from across the whole length of the spine.

Historically, experiments studying temperature responses in sensory neurons are performed with DRG neurons cultured for a few hours to overnight (*Perner and Sokol, 2021*). However, these cultures did not reflect the distribution of warm-sensitive neurons described from in vivo studies, with 26 ± 9% of all cells responding to warm temperatures, contrary to 3–10% of warm-sensitive neurons observed in vivo (*Figure 2—figure supplement 1C-E*, *Yarmolinsky et al., 2016*; *Wang et al., 2018*). We speculated that this expansion in the proportion of warm-sensitive neurons might reflect a post-injury state in which heat-sensitive neurons become sensitised to lower thermal stimuli (*Yarmolinsky et al., 2016*; *Leijon et al., 2019*). Since sensory neuron dissociation resembles an axotomy which activates injury-related pathways (*Ono et al., 2012*; *Tsujino et al., 2000*; *Zheng et al., 2007*; *Huang et al., 2012*; *Wangzhou et al., 2020*; *Nguyen et al., 2019*), we extended the commonly used DRG primary culture protocol to 3 days to allow the cells to recover from the procedure. Three-day cultures harboured approximately 6 ± 3% warm-sensitive neurons compared to overnight cultures 26 ± 9% (*Figure 2—figure supplement 1E*). Furthermore, 3-day cultures showed a reduced calcium inflow upon temperature stimulation and an improved capacity to return to baseline calcium levels upon termination of the stimulus, indicative of recovery from a post-injury state of sensory neurons (*Figure 2—figure supplement 1E–G*, *Zheng et al., 2007*; *Huang et al., 2012*). These observations are in line with data collected from in vivo calcium imaging preparations of both dorsal root and trigeminal ganglion cells in response to warm temperatures (*Yarmolinsky et al., 2016*; *Wang et al., 2018*) and suggest that 3-day cultures rather than acute/short-term preparations more accurately reflect the functional properties and abundance of warm-responsive sensory neurons that are found in behaving animals. However, whether 3-day cultures resemble native sensory neurons more closely than acute cultures in terms of their (transcriptional) identity is currently unknown.

### Trpv1 and Trpm2 deletion reduces the proportion of warmth responders

To investigate the effects of TRPV1 and TRPM2 loss on the response of sensory neurons to warm-temperature stimulation, we applied the same stimulation protocol to cultures from *Trpv1-/-* and *Trpm2-/-* animals (*Figure 2*). Lack of TRPV1 or TRPM2 led to a significant reduction in the proportion of warm-sensitive neurons compared to wildtype cultures, albeit the deletion of *Trpm2* had only a fairly

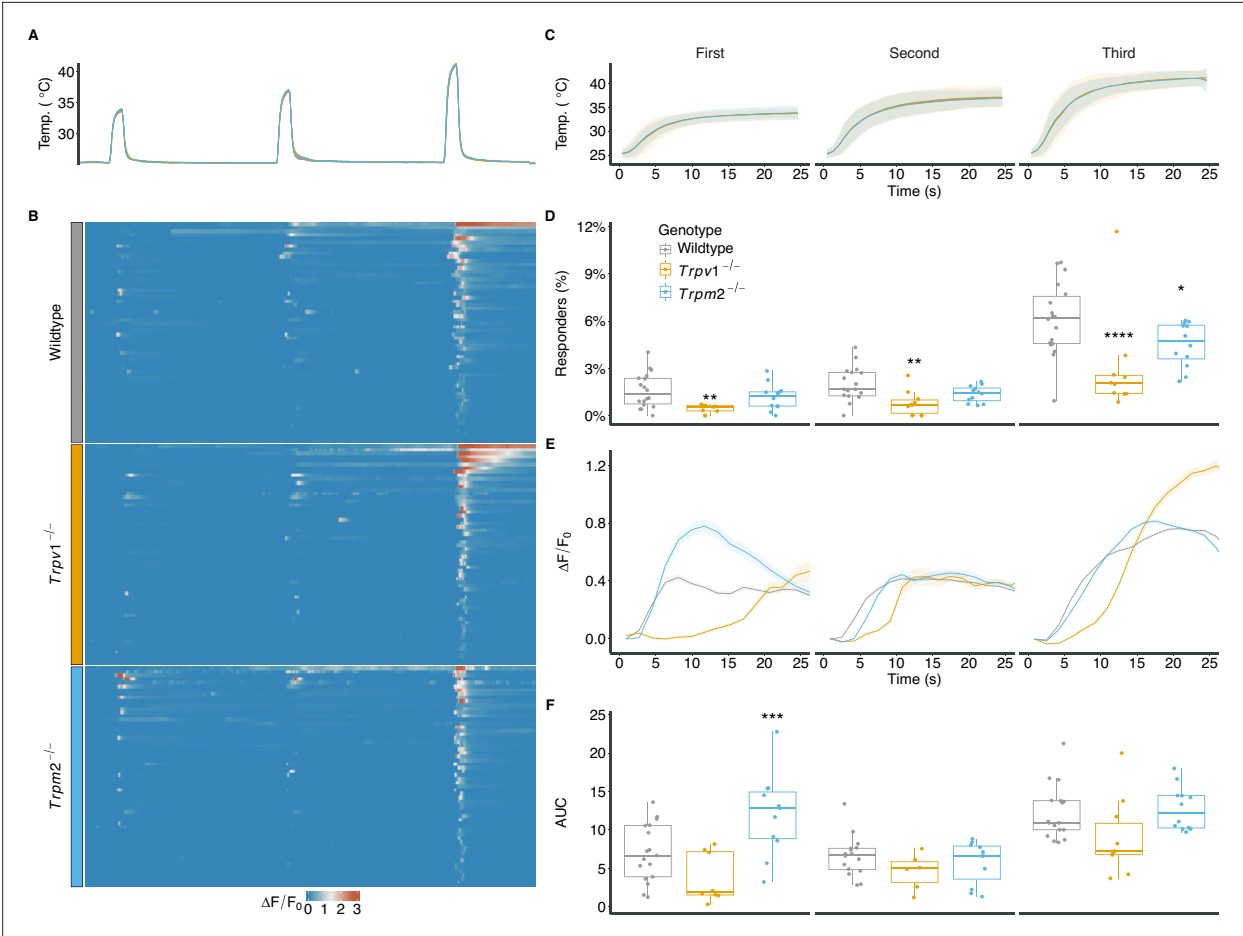

**Figure 2.** Absence of TRPV1 and to a lesser extent TRPM2 lead to a reduction in responses to warmth stimuli in dorsal root ganglion (DRG) cultures. (**A**) Experimental paradigm of temperature stimulation. Three sequential and increasing temperature stimuli of 25 seconds, with 5-minute inter-stimulus intervals. Traces represent mean temperatures for wildtype, *Trpv1*-/-, and *Trpm2*-/- cultures. (**B**) Heat map showing representative normalized ($\Delta F/F_0$) calcium response of warm-sensitive neurons (60 randomly sampled cells per genotype). (**C**) Zoom-in of the mean and SD of the three warm-temperature stimuli shown in (**A**). (**D**) The proportions of responders to each stimulus in relation to all imaged neurons from wildtype (5 animals, 18 field of views [FOVs], 6374 cells), *Trpv1*-/- (5 animals, 9 FOVs, 3009 cells), and *Trpm2*-/- (6 animals, 12 FOVs, 4315 cells). Each dot represents an FOV. ANOVA over genotype ($F(2,36) = 14.24$, $p<0.001$), stimulus ($F(1.31, 47.32) = 113.44$, $p<0.001$), and their interaction ($F(2.63, 47.32) = 9.18$, $p<0.001$). Results of post hoc multiple comparison against wildtype are indicated. (**E**) Average and SEM $\Delta F/F_0$ for all responders over the whole stimulus (as shown in **B**) from wildtype (412 cells), *Trpv1*-/- (111 cells), and *Trpm2*-/- (204 cells). (**F**) Box plots of the mean area under the curve (AUC) of ($\Delta F/F_0$) from each FOV used in (**D**). Linear mixed model over genotype and stimulus with random effects across animals and FOVs. Pairwise contrasts against wildtype are indicated. *$p<0.05$, **$p<0.01$, ***$p<0.001$.

The online version of this article includes the following figure supplement(s) for figure 2:

**Figure supplement 1.** Responder classification and comparison of culturing conditions for primary sensory neurons.

**Figure supplement 2.** Response characteristics of dorsal root ganglion (DRG) neurons to warm and hot temperatures.

small effect (*Figure 2D*, wildtype: 64 ± 2.4%; *Trpv1*-/-: 2.3 ± 1.3%; *Trpm2*-/-: 4.7 ± 1.4%). Cultures from *Trpv1*-/- animals had reduced proportions of responders across the whole range of warm-temperature stimuli (*Figure 2D*), but showed similar proportions of heat responders (neurons responding to $T ≥ 43°C$) compared to cultures obtained from wildtype animals (*Figure 2—figure supplement 2D*). In contrast to a previous study describing warm-sensitive neurons in vivo (*Yarmolinsky et al., 2016*), our *Trpv1*-/- cultures did not show a complete absence of response to warm temperatures, with some cells in the *Trpv1*-/- cultures retaining their ability to respond to warm stimuli (*Figure 2B and D*). Lack of TRPM2, on the other hand, affected the proportions of responders across both the warm and the hot temperature range (*Figure 2D*, *Figure 2—figure supplement 2D*).

Notably, warm-sensitive neurons from wildtype animals show an increase in the proportion of responders with increased stimulus intensity (second vs. first p<0.05, third vs. second p<0.0001). This was not observed in cultures from Trpv1[-/-] or Trpm2[-/-], where the increase was only significant from the second to the third stimulus. (p<0.001 and p<0.0001, respectively).

## Loss of Trpm2 alters the population response profile to warm stimuli

Next, we compared the magnitude of the responses of warm-sensitive neurons. Previous studies suggested that warm-sensitive neurons are tuned in a graded-monotonic way, that is, an increase in temperature leads to an increased response (*Wang et al., 2018*; *Yarmolinsky et al., 2016*). Wildtype and *Trpv1[-/-]* warm-sensitive neurons show an increase in response magnitude ($\Delta F/F_0$) with increasing temperature stimuli (*Figure 2E and F*). Surprisingly, warm-sensitive neurons from *Trpm2[-/-]* animals, on the other hand, respond with a significantly higher calcium inflow to the lowest temperature stimulus compared with wildtype warm-sensitive neurons. Their response to the second and third stimulus,

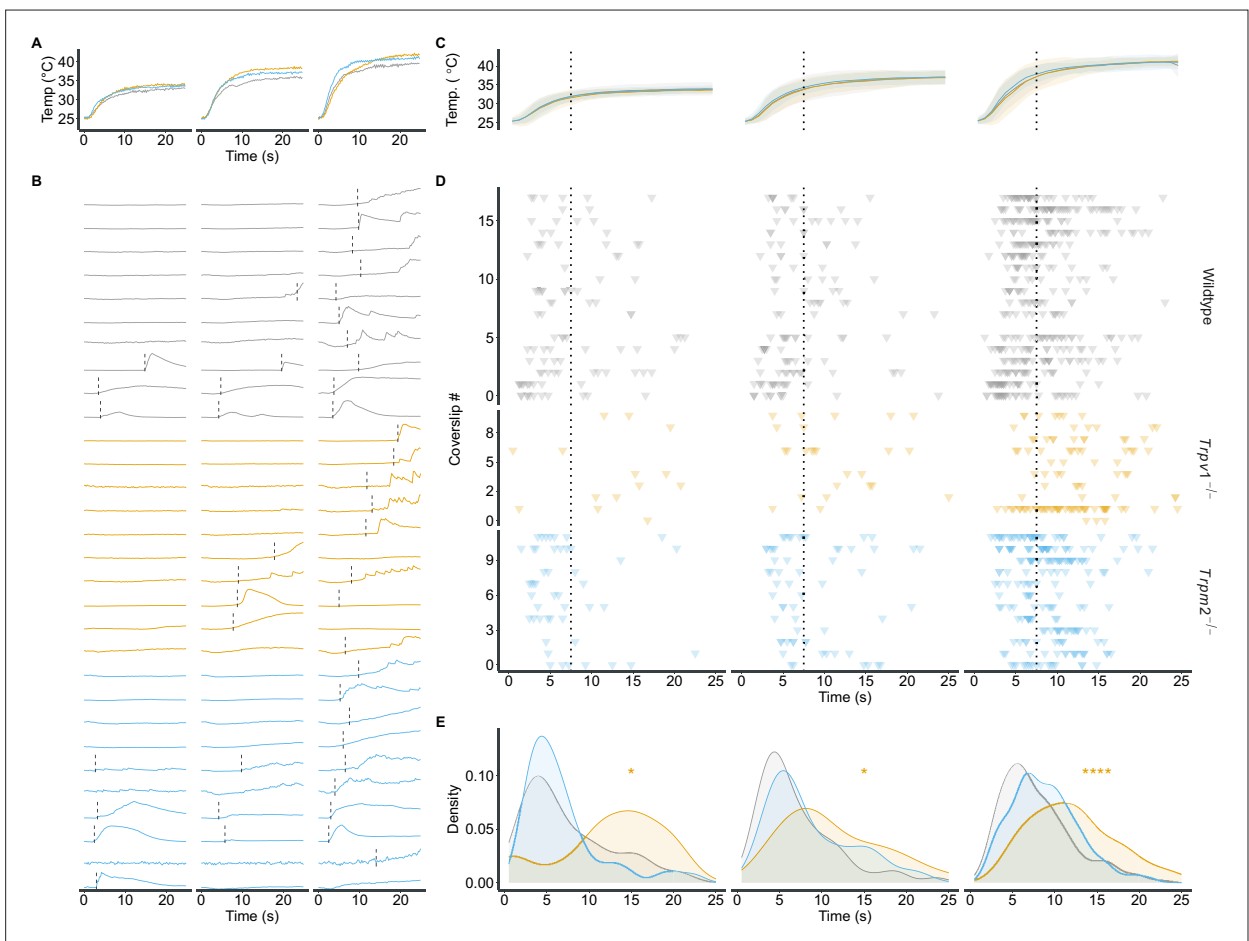

**Figure 3.** *Trpv1[-/-]* diminishes the response to dynamic temperature changes. (**A**) Temperature traces from three exemplary imaging sessions. (**B**) Individual calcium traces ($\Delta F/F_0$) of 10 representative thermosensitive neurons from each genotype in response to the applied stimuli. The position of the dashed line indicates the time when the cells exceeded 10% of their maximum $\Delta F/F_0$ during the stimulus. (**C**) Mean and SD of the three warm-temperature stimuli shown in (**A**). The dotted line indicates the separation between the dynamic and static phases, defined by the end of the peak of the smoothed temperature change rate. (**D**) Response onset of all recorded warm-sensitive neurons. Each row represents a single field of view (FOV) (see *Figure 2D*). Each triangle indicates the time point at which the individual cell responds to the stimulus as shown in (**B**). Dotted line as in (**C**). (**E**) Density plot of response time points for each genotype and stimulus. Distributions were compared against wildtype using the Wilcoxon ranked sum test with false discovery rate post hoc correction. *p<0.05, ****p<0.0001.

The online version of this article includes the following figure supplement(s) for figure 3:

**Figure supplement 1.** Responses to dynamic and static segments of warm and hot temperature stimuli.

however, are similar to wildtype warm-sensitive neurons, suggesting that tuning of the response magnitude to different warmth stimuli might be affected in $Trpm2^{-/-}$ animals.

When testing for the graded increase in response amplitude, warm-sensitive neurons from wild-type or $Trpv1^{-/-}$ only animals showed a significant increase comparing the second to the third stimulus ($p<0.0001$ and $p<0.05$, respectively). $Trpm2^{-/-}$ cells showed a significant decrease from the first to the second stimulus ($p<0.001$) followed by a significant increase from the second to third stimulus ($p<0.0001$).

## Warm-sensitive neurons vary in their response characteristics

### TRPV1 drives the dynamic phase of warm-temperature responses

Consistent with the behavioural data (*Figure 1*), the absence of TRPV1 or TRPM2 led to population changes in response to warm temperatures in primary sensory neurons. A closer look at the calcium response profiles of individual cells showed that warm-sensitive neurons also vary in when they respond to a temperature stimulus (*Figures 2B* and *3B*). To capture this variability, we computed the point at which each cell started responding to the stimulus (*Figure 3B*).

Each temperature stimulus we divided into two distinct phases: an initial, dynamic phase, in which the temperature rises rapidly. And a second, static phase, in which the temperature stabilizes (*Figure 3A*). The majority of warm-sensitive neurons from wildtype animals respond during the rising, dynamic phase of the stimulus (*Figure 3D and E*). In comparison, warm-sensitive neurons from animals lacking TRPV1 predominantly responded during the static phase of the stimulus, while $Trpm2^{-/-}$ cells did not significantly differ in their response onset from wildtype cells (*Figure 3D and E*, *Figure 3—figure supplement 1A–C*). Additionally, TRPV1-positive cells in wildtype cultures identified by their response to the TRPV1 activator capsaicin (*Caterina et al., 1997*) predominantly responded during the dynamic phase of the stimulus, compared to TRPV1-negative cells (*Figure 3—figure supplement 1D–F*). Collectively, these observations suggest that TRPV1, but not TRPM2, is involved in the response to dynamic, fast changes in temperature.

Given the strong reduction of the intracellular calcium dynamics observed in $Trpv1^{-/-}$ DRGs exposed to warm stimuli, we speculate that overexpression of $Trpv1$ would alter the response dynamics of warm-sensitive neurons, particularly during the rising phase of the stimulus. To test this hypothesis, we made use of a previously described animal model which overexpresses $Trpv1$ in TRPV1-positive cells ($Trpv1$-OX, *Hanack et al., 2015*). warm-sensitive neurons from $Trpv1$-OX animals showed a significantly higher propensity to respond during the dynamic phase of the stimulus compared to wildtype cultures (*Figure 4A–E*). These results align with our previous observations and further suggest that TRPV1 abundance directly regulates the onset and speed of a temperature response. Notably, DRG cultures from $Trpv1$-OX animals showed nearly double the proportion of warm-sensitive neurons compared to wildtype cultures (wildtype: 6.8 ± 3.9%; $Trpv1$-OX: 12.8 ± 0.6%), which suggests that TRPV1-overexpression reduces the response threshold of warm-sensitive neurons.

Does the enrichment of cells responding during the dynamic stimulus phase affect the behaviour of the animals in the CPT? Indeed, $Trpv1$ overexpression led to a significantly stronger avoidance of the 38°C side in the CPT (*Figure 4F and G*). Interestingly, $Trpv1$-OX animals crossed significantly less between chambers, compared to wildtype animals, while having similar duration of stays at the test chamber (*Figure 4H–J*), suggesting that $Trpv1$-OX animals discriminate temperatures more rapidly.

## A drift-diffusion model uncovers differences in evidence accumulation across genotypes

In the previous sections, we have detailed the distinct effects that TRPV1 and, to a lesser extent, TRPM2 exert on the temperature responses of sensory neurons. While these cellular-level findings are illuminating, they present a challenge when it comes to directly relating them to the more complex, multifaceted behaviours observed in our temperature preference assay. To bridge this gap and extract parameters that could be directly correlated with the neuronal data, we conceptualized the temperature preference assay as a continuous decision-making process (*Figure 5A and B*), allowing the use of established evidence accumulation frameworks. These models have been shown to successfully recapitulate animal and human behaviour in sensory decision tasks involving different modalities (*Lebovich et al., 2021*; *Hanks et al., 2015*; *Stine et al., 2023*) and have even been directly linked to neural observations (*Deco et al., 2013*; *Gordon et al., 1998*; *Gupta et al., 2021*).

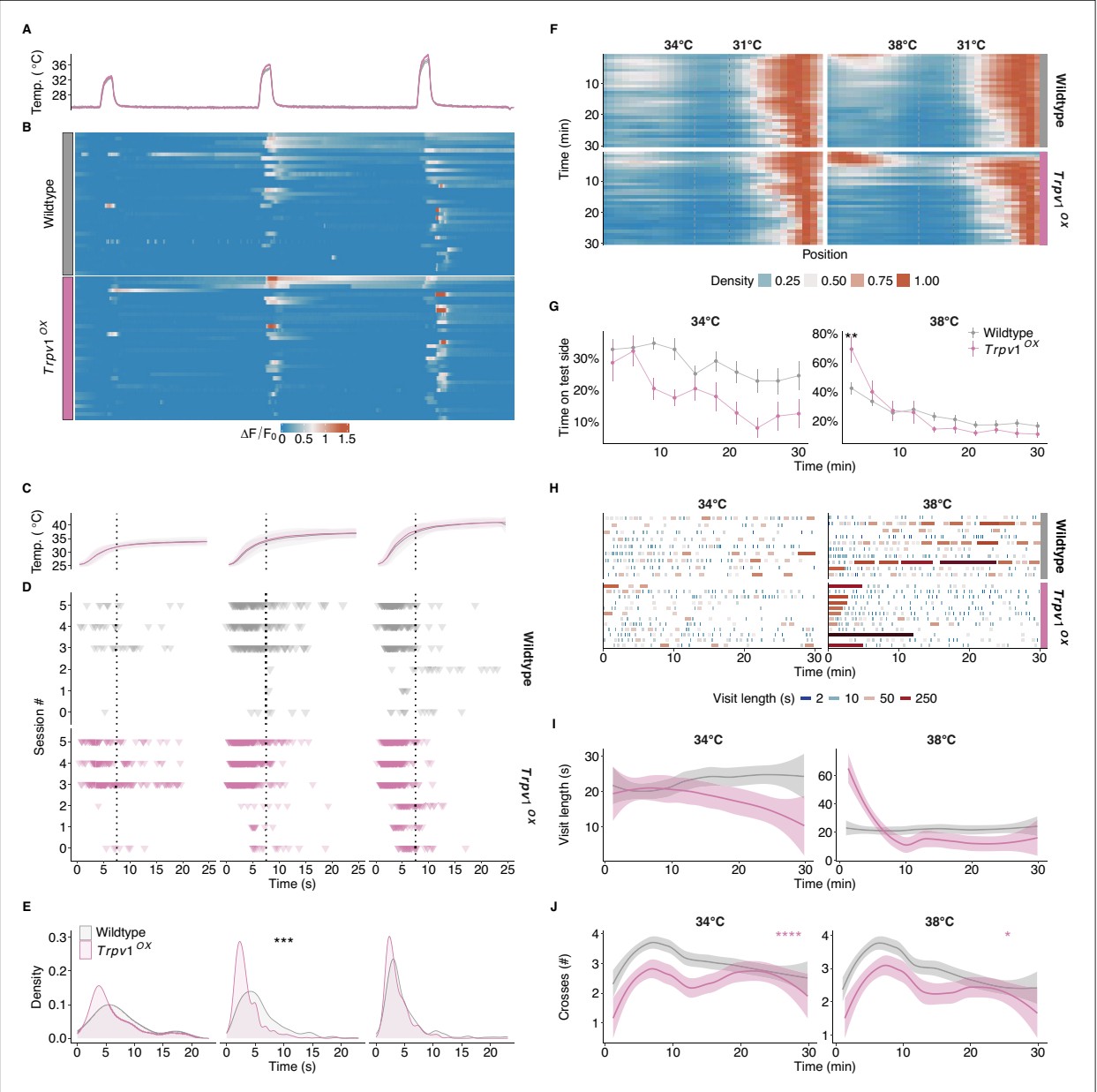

**Figure 4.** High TRPV1 expression levels promote dynamic warm-temperature detection and enhance temperature preference. (**A**) Mean temperatures from all experiments and imaging sessions for wildtype (3 animals, 6 field of views [FOVs], 3133 cells) and *Trpv1-OX* (2 animals, 5 FOVs, 3754 cells) cultures. (**B**) Examples of normalized ($\Delta F/F_0$) calcium responses of warm-sensitive neurons responding to any of the stimuli depicted in (**A**). 42 randomly sampled cells from each genotype. (**C**) Mean and SD of the three warm-temperature stimuli applied. The dotted line indicates the separation between the dynamic phase and the static phase (see *Figure 3C*). (**D**) Response initiation of all warm-sensitive neurons imaged from wildtype and *Trpv1-OX* animals. Each row represents an individual imaging session. Each triangle denotes the time point at which the cell responds to the stimulus, as shown in *Figure 3B*. Dotted line as in (**C**). (**E**) Density representation of response time points for each genotype and stimulus. Distributions were compared against wildtype using the Wilcoxon ranked sum test. (**F**) Density maps of all wildtype (n = 48) and *Trpv1-OX* (n = 12) animals in the chamber preference test (CPT) over time. (**G**) Mean proportion and SEM of time spent in the test chamber for animals shown in (**F**) over time, binned in 3-minute intervals. ANOVA over genotype (34°C: F(1,51) = 6.49, p<0.05), time (34°C: F(6.12,311.93) = 4.49, p<0.001; 38°C: F(4.71,240.22) = 28.27, p<0.001), and their interaction (38°C: F(4.71,240.22) = 4.42, p<0.001). Results of post hoc multiple comparison by timepoint against wildtype are indicated. (**H**) Overview of the frequency and length of the visits to the test chamber for all animals shown in (**F**). Each visit is coloured by the log2 of its length to highlight varying visit lengths. (**I**) Averaged and smoothed visit lengths in a 3-minute rolling window with a 1-minute lag. The shaded area depicts the 95% confidence interval. Linear mixed model over genotype and time with random effects across animals. (**J**) As in (**I**) but for the number of crosses. Cox regression over genotype (34°C: X² (2) = 17.52, p<0.001; 38°C: X² (2) = 4.13, p<0.05). *p<0.05, **p<0.01, ***p<0.001, ****p<0.0001.

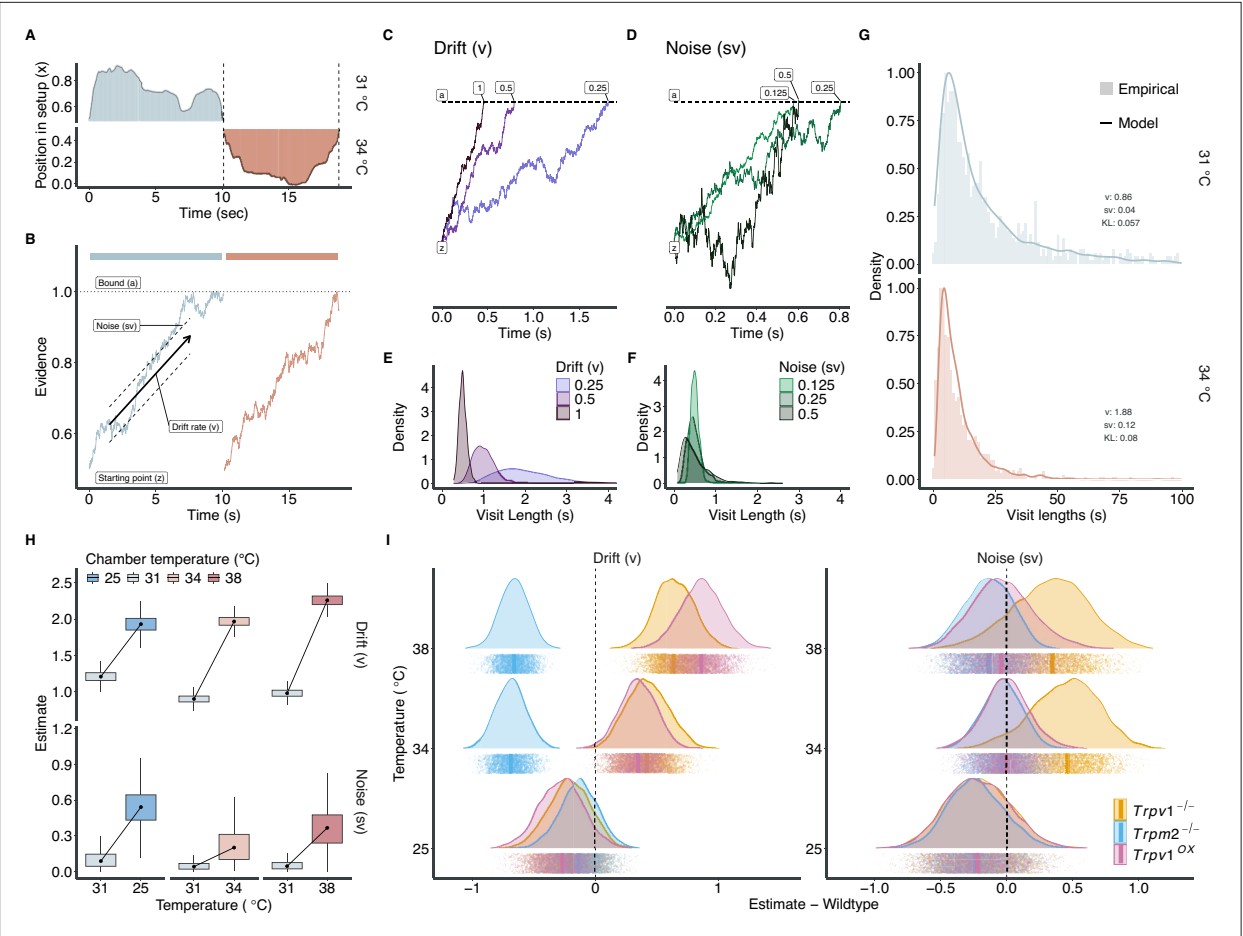

**Figure 5.** Modelling the varying roles of TRPV1 and TRPM2 on warm-temperature detection. (**A**) Two example episodes of an animal inside the chamber preference test (CPT), crossing from one chamber to the other (see *Figure 1G*). Dashed line represents crossing time points between chambers. (**B**) Examples of possible evidence accumulation process for the two episodes in (**A**) using a drift diffusion model (DDM). (**C, D**) Simulations of a drift diffusion process with fixed starting points (z = 0.5) and bound (a = 1) while varying drift rates (v in **C**) and noise (sv in **D**). (**E**) and (**F**) depict the resulting distributions of visit lengths when simulating 1000 trials with the parameters from (**C**) and (**D**), respectively. (**G**) Distributions of visit lengths at 31°C vs. 34°C in wildtype animals. Insets show the estimated parameters for v and sv at each temperature and the Kullback–Leibler (KL) divergence between the model (continuous density line) and the empirical data (histogram). See *Figure 5—figure supplement 1* for all model fits. (**H**) Box plots of drift v and noise sv estimates for both neutral (31°C:, solid line) and test (25–38°C, dashed line) chambers for wildtype animals resulting from hierarchical Markov chain Monte Carlo (MCMC) sampling. (**I**) Neutral chamber (31°C) corrected and wildtype-subtracted estimates of drift and noise for all genotypes. The dashed line represents the wildtype reference. Points indicate individual MCMC samples, and vertical lines the median of each distribution.

The online version of this article includes the following figure supplement(s) for figure 5:

**Figure supplement 1.** Model fits and drift diffusion model (DDM) simulations for all genotypes and test temperatures.

In our experimental setup, an animal enters a chamber and begins to accumulate evidence (i.e. it continuously collects and computes spatial temperature information) that drives its decision to stay or leave the chamber (*Figure 5A*). The resulting time spent in each chamber varies from visit to visit (*Figures 1G and 5A*), highlighting the need to account for a dynamic and stochastic decision-making process. This stochastic variability could be due to many factors, such as variability between animals, fluctuations in cognitive variables (e.g. attention or motivation), different exploration speeds of the mice (which alters the duration the mice are exposed to different temperatures, and which could result in a perceived change in perceptual threshold) or actual changes in perceptual threshold over time. Traditional approaches to analysing these behaviours have focused on final outcomes or mean stay time and thus may have overlooked these more subtle nuances (*Vandewauw et al., 2018*; *Tan and McNaughton, 2016*; *Pogorzala et al., 2013*; *Bautista et al., 2007*), without considering

the fluctuating nature of sensory perception and the variability of decision-making across visits and between individual animals.

## A drift diffusion model recapitulates the animals behaviour in the CPT

To model this decision-making process, we opted for a drift diffusion model (DDM; *Ratcliff, 1978*). This model provides a quantitative framework to delve into how animals integrate sensory information over time, leading to their decision to stay or leave a chamber (*Figure 5*).

When an animal enters a chamber, it starts at a certain point ($z$) from which the accumulation of sensory information/evidence begins (*Figure 5B*). Over time, and while the temperature information is integrated, the information is accumulated at a certain rate (drift rate, $v$) towards a decision point (decision bound, $a$). Additionally, the drift rate is also allowed to fluctuate (noise, $sv$), which represents the variability of information accumulation (sensory perception). This process evolves until the decision bound ($a$) is reached, which prompts the animal to leave the chamber. *Figure 5C and D* show simulations of the DDM, which allow an intuition into how varying levels of drift and noise (while keeping the starting point and decision bound fixed) alters the distribution of visit lengths throughout an experiment (*Figure 5E and F*). Higher rates of drift (with fixed noise levels) lead to overall shorter visit lengths (*Figure 5E*). High noise levels (with fixed drift rates) also lead to shorter visit lengths, albeit with larger variability (*Figure 5F*).

We fit a model with varying drift rate and noise onto the visit length distributions of wildtype animals recorded for each temperature combination in the CPT (*Figure 5G*, *Figure 5—figure supplement 1*). The model suggests higher drift rates at temperatures below and above 31°C in wildtype animals (*Figure 5H*). This means that the sensory information to leave the non-neutral chambers is accumulated faster. The model also suggests an increase in noise in the test chambers, hinting that the evidence accumulation at 31°C is particularly stable (*Figure 5H*). Both observations are in line with the development of preference for the 31°C chamber in the varying temperature conditions.

## Varying effects of TRPM2 and TRPV1 on evidence accumulation

We fit the model onto all behavioural data collected in this study (*Figure 5I*, *Figure 5—figure supplement 1*). For animals lacking TRPM2, the model yields a lower drift rate at 34°C and 38°C compared to wildtypes (*Figure 5I*). This suggests that loss of TRPM2 leads to a slower evidence accumulation at warm temperatures, reflecting an overall failure to develop a preference for 31°C throughout the experiment (*Figure 1*). In *Trpv1*[-/-] animals, on the other hand, we observed higher drift rates as well as higher noise levels in the warmer (34°C and 38°C) chambers (*Figure 5I*). These findings suggest that *Trpv1*[-/-] animals accumulate environmental temperature evidence faster than wildtype animals, but the fidelity of the thermal inputs is compromised. We speculate that the balance of these two variables might lead to a similar overall preference development compared to wildtype animals (*Figure 1*). Interestingly, the overexpression of TRPV1 also leads to an increased drift rate at warm temperatures, albeit with a similar noise level to wildtypes. This combination leads to greater avoidance of 34°C and 38°C (as observed in *Figure 4*).

Notably, all genotypes show similar drift and noise estimates at 25°C, consistent with their behavioural preference (*Figure 1—figure supplement 2*), suggesting that TRPV1 and TRPM2 mainly control responses to warm temperatures (*Figure 5I*). In summary, we find that the DDM successfully parametrizes the behavioural data obtained from the CPT. Furthermore, the model allowed a more in-depth insight into how the loss of either TRPV1 or TRPM2 differentially alters the detection of warm temperatures, highlighting their importance in behavioural adaptation to innocuous temperatures.

## Discussion

Environmental temperatures are detected by sensory nerve fibres innervating the skin. The mechanisms behind warm-temperature detection have recently gained increased attention, with three ion channels – TRPV1, TRPM2, and TRPM8 – as the main candidates (*Tan and McNaughton, 2016*; *Yarmolinsky et al., 2016*; *Paricio-Montesinos et al., 2020*). In this study, we developed a novel temperature preference assay, integrating ambient and floor temperatures, to investigate the roles of TRPV1 and TRPM2 in temperature detection. Our results reveal distinct behavioural responses to warm temperatures mediated by these channels. Applying a modelling framework to the animals

behaviour, we observed unique deficits in TRPV1 and TRPM2 knockout animals compared to wild-type mice. On the cellular level, the loss of either TRPV1 and, to a lesser extent, TRPM2 resulted in a decreased proportion of warm-sensitive neurons, with TRPV1 playing a pivotal role in detecting rapid, dynamic temperature changes, while TRPM2 loss appeared to affect the population response of warm-sensitive neurons.

## Behavioural analysis in temperature preference assays

The introduced chamber preference assay, integrating both ambient and floor temperatures, improves on the conventional temperature preference assays. Notably, at 34°C, a temperature that is close to the thermoneutral 31°C, animals demonstrated a clear avoidance of the 34°C side in the CPT, but failed to do so in the conventional TPP assay (*Figure 1—figure supplement 1I*). This preference underscores the importance of integrating multiple sensory inputs such as ambient air and contact temperatures in forming a coherent thermal perception, a complexity often overlooked in simpler thermal assays.

Consistent with previous findings, our results reveal that the absence of TRPM2 impedes the development of a preference for warmer temperatures (*Tan and McNaughton, 2016*). In contrast, animals lacking TRPV1 exhibited behavioural patterns similar to their wildtype counterparts, spending comparable amounts of total time in the warm chamber (*Figure 1E and F*).

Intriguingly, a finer characterization of the dynamics of the animal behaviour in the assay revealed differences between *Trpv1*$^{-/-}$ and *Trpm2*$^{-/-}$ animals, particularly in the frequency of crossings between chambers and the time spent in each chamber (*Figure 1G–J*). These behavioural nuances were further elucidated by modelling the behaviour with an evidence-accumulation model (*Figure 5*, *Figure 5—figure supplement 1*). This model, a novel approach for such behavioural assays in general and for temperature as a sensory modality in particular, uncovered an impaired process of evidence accumulation within the warm chambers in *Trpm2*$^{-/-}$ animals. Moreover, we could explain the more frequent chamber crossings of *Trpv1*$^{-/-}$ animals by the fact that they accumulated evidence (information of preferred temperature) more error-prone and thus erratically (*Figure 5I*).

## Cellular insights into warm-temperature sensation

Cultures from *Trpv1*$^{-/-}$ animals exhibited a substantial decrease in the proportion of warm-sensitive neurons (*Figure 2*). This is similar to previous studies from trigeminal neurons, where the loss of TRPV1 led to a complete loss of warm-temperature responses (*Yarmolinsky et al., 2016*). While the role of TRPV1 was more salient in the warm-temperature range, *Trpm2* knockouts displayed a reduction in temperature responsiveness across a broader spectrum, extending into hotter temperatures, albeit the overall effect *Trpm2*-deletion had on temperature responses in DRG cultures appeared very subtle (*Figure 2*, *Figure 2—figure supplement 2*). This is in line with previous studies highlighting only a subtle loss in warm-/heat-responsiveness in DRG cells of *Trpm2*$^{-/-}$ animals (*Tan and McNaughton, 2016*; *Vilar et al., 2020*; *Mulier et al., 2020*).

Interestingly, in initial experiments using DRG neurons from *Trpm2*$^{-/-}$ and *Trpv1*$^{-/-}$ animals cultured overnight, we failed to reproduce the previously reported reduction in warm-sensitive neurons in *Trpm2*$^{-/-}$ and *Trpv1*$^{-/-}$ sensory neurons (*Figure 2—figure supplement 1H*, *Tan and McNaughton, 2016*; *Yarmolinsky et al., 2016*). The inability to reproduce the aforementioned cellular phenotypes in cultured sensory neurons might be due to two factors: the abundance of warm-sensitive neurons and the variability in their proportions between experiments and animals (*Figure 2*, *Figure 2—figure supplement 2*). These require a larger sampling of sensory neurons from multiple animals for a reliable estimation of effects, something that is often lacking in previous studies of cellular warm-temperature detection (*Tan and McNaughton, 2016*). Furthermore, overnight cultures, which are the de facto standard in the field, might be more akin to an injury model (*Figure 2—figure supplement 1*). The 3-day cultures presented in this study allow the cells the time to partially regenerate from the harsh dissociation procedure (*Renthal et al., 2020*) and pose an alternative that more closely resembles the physiological condition.

## TRPV1: Bridging cellular data with behavioural patterns

The cellular data predict that animals lacking TRPV1 would have large deficits in their ability to detect warm temperatures. Yet, overall, *Trpv1*$^{-/-}$ animals stay in the thermoneutral chamber for a similar

proportion of time as wildtype controls. Analysis of the remaining warm-sensitive neurons in *Trpv1*<sup>-/-</sup> animals revealed a critical insight: these neurons predominantly respond during the static phase of the temperature stimuli.

In the CPT, animals frequently transition between chambers, experiencing rapid temperature changes upon crossing, but then spend most of their time in an isothermal environment (*Figure 1—figure supplement 1H*). This suggests that *Trpv1*<sup>-/-</sup> animals primarily rely on static temperature information for thermal detection, rather than rapidly fluctuating temperatures. This is reflected in the higher number of crossings between the different thermal chambers, coupled with shorter visits to the hotter chambers. This set of results led us to hypothesize that rapidly changing thermal information perceived during the transitions of the animals is not properly detected by *Trpv1*<sup>-/-</sup> animals.

This hypothesis is further supported by the behavioural model (*Figure 5*). It indicates that *Trpv1*<sup>-/-</sup> animals exhibit a higher drift rate in warmer test chambers, suggesting an avoidance of these temperatures. However, the increased noise in the DDM points to a less reliable temperature detection mechanism. Although noise in DDMs can encompass various sources of variability – ranging from peripheral sensory processing to central mechanisms like attention or motor initiation – the most parsimonious interpretation in our study aligns with a perceptual deficit, given the altered temperature-responsive neuronal populations we observed. This implies that, despite the substantial loss of warm-sensitive neurons, the remaining neuronal population provides sufficient information for the detection of warmer temperatures, albeit with reduced precision.

The reduced precision might stem from the loss of dynamic temperature responders (*Figure 3*). These warm-sensitive neurons might be crucial in detecting a rapid change of temperature (e.g. when the animals move across different thermal environments). This is highlighted by findings from TRPV1-overexpressing animals. These animals, equipped with an enhanced ability to respond to dynamic temperature changes (*Figure 4*), have a higher drift rate and lower noise levels in warmer chambers in the model (*Figure 5*). These characteristics lead to a faster and more precise choice in the CPT. Collectively, these results highlight the direct role of TRPV1, and its expression levels, in the precise temporal detection of warm temperatures. This could also explain the consistent albeit subtle involvement of TRPV1 in operant assays of temperature perception where temperature stimuli are applied rapidly (*Yarmolinsky et al., 2016*; *Paricio-Montesinos et al., 2020*).

## TRPM2: Cellular mechanisms and behavioural implications

A reversed scenario unfolds for TRPM2. The behavioural data suggests a strong deficit in detecting warm temperatures. In *Trpm2*<sup>-/-</sup> cultures, warm-sensitive neurons appear less abundant compared to wildtype DRG cultures, albeit the contribution of *Trpm2* appears to be less robust compared to that of *Trpv1* (*Figure 2*). *Trpm2*<sup>-/-</sup> warm-sensitive neurons also do not differ from wildtype warm-sensitive neurons in their response timings (*Figure 3*). It is tempting to hypothesize, given the results from *Trpv1*<sup>-/-</sup>-cultures, that TRPM2 affects the static phase of temperature detection. We conducted various analyses on the static responses of *Trpm2*<sup>-/-</sup> warm-sensitive neurons (data not shown), but failed to uncover significant differences to wildtype warm-sensitive neurons.

The only difference we found when comparing *Trpm2*<sup>-/-</sup> cultures to warm-sensitive neurons from wildtype and *Trpv1*<sup>-/-</sup> animals was their increased response magnitude to the first stimulus, without affecting the subsequent stimuli (*Figure 2*). Previous studies hypothesized that peripheral warm and hot temperature perception requires a combination of population and graded rate coding (*Yarmolinsky et al., 2016*; *Wang et al., 2018*). This means that a series of temperature stimuli with increasing intensity should activate and recruit more neurons, as well as increase their response magnitude. *Trpm2*<sup>-/-</sup> warm-sensitive neurons deviate from this model by having a U-shaped response magnitude to increasing temperature stimuli (*Figure 2F*). Two different temperatures leading to similar response magnitudes could impair the animals' ability to differentiate these two temperatures and thereby lead to the observed phenotypes in the CPT (*Figures 1 and 5*). The question remains as to why the loss of TRPM2 would lead to an increase in the response magnitude, and whether this relatively small effect would have any bearing on temperature sensing in vivo.

Given the relatively minor cellular phenotype observed by TRPM2 deletion, there are several alternative hypotheses that could explain the discrepancy between the behavioural and cellular phenotypes.

It is possible that permanent deletion of TRPM2 in the *Trpm2*[-/-] mice results in developmental defects and/or compensatory mechanisms that mask a more prominent phenotype that might occur if the channel would be acutely blocked. In the absence of specific TRPM2 antagonists, such an experiment is currently not possible. Another explanation involves the specific population of warm-sensitive neurons reliant on TRPM2. Single-cell sequencing and functional analyses of DRG suggest that warm-sensitive neurons form two, genetically distinct populations, one of which prominently expresses *Trpm2* mRNA transcripts (*Sharma et al., 2020*; *Qi et al., 2023*). Different genetic/functional populations of sensory neurons often show diverging spinal innervation, with different upstream processing pathways (*Marmigère and Ernfors, 2007*; *Wu et al., 2021*; *Choi et al., 2020*; *Qi et al., 2023*). The genetic separation could hint at multiple neural innervation routes for innocuous temperature information. In this context, the loss of TRPM2 might specifically impair warm-temperature perception in thermoregulation-specific innervation pathways, without significantly affecting perceptual temperature discrimination performance (*Paricio-Montesinos et al., 2020*).

A likely third possibility implies that the role of TRPM2 in temperature detection extends beyond its expression in sensory neurons. Particularly, it is possible that TRPM2 mediates part of its effect via the hypothalamic preoptic area (POA), a temperature-sensitive brain region involved in body temperature regulation (*Morrison and Nakamura, 2019*). Preoptic TRPM2 has been shown to mediate autonomic thermoregulatory responses upon warm-temperature stimulation (*Song et al., 2016*; *Kamm et al., 2021*; *Yang et al., 2023*). Preoptic temperature pathways may not only drive autonomic thermoregulatory responses, but can also influence temperature preference behaviour (*Morrison and Nakamura, 2019*; *Tan et al., 2016*). Considering that, within the time frame of the CPT, ambient temperature changes are directly transferred to the POA, it is possible – but not yet tested – that preoptic TRPM2 is involved in the choice of comfort temperature. The use of conditional Trpm2 knockout animals may help to clarify this aspect in future studies.

On a more general note and considering the choice of the behavioural assay used in different studies, either an operant task or a preference assay without training of the mice: depending on the task the animals perform, likely requires qualitatively and quantitatively different thermal inputs, thereby possibly explaining the different phenotypes observed in individual TRP channel knockout mouse models.

## Implications of the findings

This study introduces an alternative protocol to culture DRG neurons to reduce their (thermal) hypersensitivity, an innovative behavioural assay, and methodologies for analysing animal behaviour in temperature preference assays. We emphasize the importance of examining the dynamics of perceptual decision-making and incorporating behavioural modelling. Significantly, we demonstrate that TRPV1 and TRPM2 channels contribute differently to temperature detection, supported by behavioural and cellular data. This research not only advances our understanding of thermal perception mechanisms but also adds new dimensions for integrating cellular and behavioural data to study the neural foundations of temperature sensation.

## Limitations of the study

In this study, only male animals were used to study temperature preference in the CPT assay. It is possible that sex differences modulate thermal preference in a TRP channel-dependent manner, as previously suggested (*Carstens et al., 2024*). A systematic comparison of TRP-channel involvement in sex differences in thermal detection across different thermal modalities (operant-based choice assays, thermal preference chamber test and others) awaits future analysis. The 3-day DRG cultures used in this study display a response profile that is more similar to the (rare) warmth responses detected in vivo compared to acute DRG cultures (which display exaggerated/sensitized warmth/heat responses). However, longer culturing times may result in cellular changes and a drift of neuronal identity away from their native state. Future in vivo DRG

recording/imaging studies using TRP-channel knockout mouse models will help to reveal how warmth responses are coded in the native cellular setting.

# Methods

## Key resources table

| Reagent type (species) or resource | Designation | Source or reference | Identifiers | Additional information |
|---|---|---|---|---|
| Strain, strain background (*Mus musculus*) | Wildtype | Janvier Laboratories | RRID:IMSR_RJ:C57BL-6NRJ | C57BL/6NRj |
| Genetic reagent (*M. musculus*) | *Trpm2*[-/-] | Yasuo Mori | RRID:MGI:5697655 | B6;Trpm2[tm1Yamo]/Uhg |
| Genetic reagent (*M. musculus*) | *Trpv1*[-/-] | David Julius | RRID:IMSR_JAX:003770 | B6.129X1-Trpv1[tm1Jul]/J |
| Genetic reagent (*M. musculus*) | *Trpv1-OX* | Interfacultary Biomedical Faculty, University of Heidelberg | RRID:IMSR_JAX:027390 | C57BL/6N-Tg(Trpv1)5917Jsmn/J |
| Genetic reagent (*M. musculus*) | *Trpv1*[cre] | The Jackson Laboratory | RRID:IMSR_JAX:017769 | B6.129-Trpv1[tm1(cre)Bbm]/J |
| Genetic reagent (*M. musculus*) | Rosa-DTA | The Jackson Laboratory | RRID:IMSR_JAX:006331 | Gt(ROSA)26[Sortm1(DTA)Jpmb]/J |
| Genetic reagent (*M. musculus*) | *Trpv1-Abl* | This paper | | F1 from crossing *Trpv1*[cre] and RosaDTA |
| Genetic reagent (*M. musculus*) | *Trpm8*[-/-] | David Julius | RRID:IMSR_JAX:008198 | Trpm8[tm1Jul] |
| Chemical compound, drug | 4-(2-hydroxyethyl)–1-piperazineethanesulfonic acid (HEPES) | Carl Roth | Cat# 9105.4 | |
| Chemical compound, drug | Cal-520 AM | AAT Bioquest | Cat# 21130 | |
| Chemical compound, drug | Calcium chloride dihydrate | MerckMillipore | Cat# 1023821000 | |
| Chemical compound, drug | Capsaicin | Tocris | Cat# 462 | |
| Chemical compound, drug | Isoflurane | Baxter | Cat# HDG9623 | |
| Chemical compound, drug | Magnesium chloride | Sigma-Aldrich | Cat# M8266 | |
| Chemical compound, drug | Pluronic F127 | Invitrogen | Cat# P6866 | |
| Chemical compound, drug | Poly-D-lysine (PDL) | Sigma | Cat# P7886 | |
| Chemical compound, drug | Potassium chloride | Labochem International | Cat# LC-5916.1 | |
| Chemical compound, drug | Proteinase K | Carl Roth | Cat# 7528.1 | |
| Chemical compound, drug | Sodium chloride | Sigma-Aldrich | Cat# 31434 | |
| Chemical compound, drug | Tris-HCl | Carl Roth | Cat# 5429.3 | |
| Chemical compound, drug | Trypsin-EDTA 0.05% | Thermo Fisher | Cat# 25300054 | |
| Chemical compound, drug | Antibiotic-Antimitotic (100×) | Thermo Fisher Scientific | Cat# 15240062 | |
| Peptide, recombinant protein | L-alanyl-L-glutamine dipeptide (GlutaMAX) | Invitrogen | Cat# 35050 | |

*Continued on next page*

*Continued*

| Reagent type (species) or resource | Designation | Source or reference | Identifiers | Additional information |
|---|---|---|---|---|
| Peptide, recombinant protein | Laminin | Sigma | Cat# L2020 | |
| Peptide, recombinant protein | Collagenase | Sigma | Cat# C0130 | |
| Peptide, recombinant protein | Bovine serum albumin (BSA) fraction V | Carl Roth | Cat# T844.1 | |
| Biological sample (Bos taurus) | Fetal calf serum (FCS) – EU Approved | Invitrogen | Cat# 10270 | |
| Other | Dulbecco's PBS | Thermo Fisher Scientific | Cat# 14040141 | Sterile, commercial phosphate-buffered saline for cell-culture use |
| Other | DMEM/F12 without Glutamine | Thermo Fisher Scientific | Cat# 21331046 | Sterile, commercial medium for cell-culture |
| Software, algorithm | LOGO! Soft Comfort | Siemens | | |
| Software, algorithm | MetaFluor | Molecular Devices | RRID:SCR_014294 | |
| Software, algorithm | Miniscope DAQ | UCLA Miniscope Team | RRID:SCR_021480 | |
| Software, algorithm | Python 3.10 | Python Software Foundation | RRID:SCR_008394 | |
| Software, algorithm | R 4.3.2 | R Core Team | RRID:SCR_001905 | |
| Software, algorithm | FFmpeg 4.2 | FFmpeg Developers | RRID:SCR_016075 | |
| Software, algorithm | Thermes USB DAQ | Physitemp | | |
| Software, algorithm | HSSM 0.1.5 | GitHub | RRID:SCR_026356 | |
| Software, algorithm | MINIROCKET (sktime 0.35.0) | *Dempster et al., 2021*; *Dempster and Jafferji, 2022* | | https://doi.org/10.48550/arXiv.2012.08791 |
| Software, algorithm | Suite2p | GitHub | RRID:SCR_016434 | |
| Software, algorithm | Cellpose | GitHub | RRID:SCR_021716 | |

## Animals and housing

All animal care and experimental procedures were approved by the local council (Regierungsprsidium Karlsruhe, Germany) under protocol numbers G-201/16 and T05-19. Animals were kept under specific-pathogen-free (SPF) conditions and a 12-hour day-night cycle. Housing temperature and humidity were maintained at 22 ± 2°C and 50–60%, respectively. Animals were fed ad libitum with Altromin Rod 16 or Rod 18 animal food. The housing environment was enriched using Crincklets Nest-Pads and ABBEDD LT-E-001 bedding. For this study, only male animals were used as we aimed to compare our results with previous studies which exclusively used male animals (*Yarmolinsky et al., 2016*; *Tan and McNaughton, 2016*). Mice between 6 and 25 weeks of age were used for the experiments.

## Thermal preference chamber design, operation, and video capture

The thermal preference chamber consisted of two expanded polystyrene boxes connected using plastic glue and sealed with silicone (*Figure 1—figure supplement 1A*) with dimensions of 26.1 cm × 60.3 cm × 19.8 cm (w × l × h). The 4.3-cm-thick Styrofoam walls provided the necessary thermal insulation for the experiments. Inside the enclosure, the animals movement was limited by a 13 cm × 31.4 cm × 15.5 cm steel cage placed on top of a stainless-steel baseplate. To create two thermally isolated chambers, the enclosure, cage, and baseplate were adjusted to form a 4.3 cm × 6.3 cm × 6.3 cm tunnel. The baseplate and cage were custom-built by our institute's mechanical workshop. The cover combined foam and wood as insulators, with an acrylic glass inset for observation purposes.

Temperature within the chamber was regulated by two Peltier elements attached to heat sinks, each connected to a generic computer fan for efficient temperature distribution. To avoid over-heating, the Peltier elements were connected to a Multitemp III circulating water pump (Amersham Biosciences) set to 28 °C. These elements were managed by a modified Siemens LOGO TD! controller, programmed for precise temperature adjustments, and accessed using the LOGO! Soft Comfort (Siemens) software. For monitoring, two Physitemp IT-18 flexible thermocouples were attached to the

chamber walls, serving as reference thermometers. Data capture was conducted using a customer-grade webcam (Spedal), linked to the UCLA miniscope projects capture software, operating at 20–30 Hz. The output files from each recording session were concatenated using *ffmpeg* software for subsequent processing. For control experiments using the two-plate temperature preference test (*Figure 1—figure supplement 1*), we used the BIO-T2CT system by BioSeb with the same camera setup as described above.

## Thermal preference tests, processing, and analysis

Animals were transported to and acclimatized in the experimental room for at least 24 hours before the experiments. The room maintained a dim light setting and a 12-hour day-night cycle. To ensure temperature stability, the setup was allowed to stabilize for 90 minutes before starting an experiment (*Figure 1—figure supplement 1B–D*). Once the setup reached a stable temperature, the lid was briefly opened (*Figure 1—figure supplement 1*), and an animal was placed into the enclosure, the lid closed again, and allowed to roam freely for at least 30 minutes, before replacing it with the next animal.

Due to the dim light conditions and the reflections on the cover of the enclosure (see *Figure 1* for an example), conventional animal tracking approaches that rely on image contrast failed to provide robust outputs (data not shown). Therefore, we employed a neural-network based approach, namely *DeepLabCut*, for tracking the animals in the setup (*Mathis et al., 2018*). For this purpose, we trained a ResNet50 network to track specific points on the animal: the snout, right ear, left ear, body centre, tail base, and tail tip (*Figure 1*). Additionally, we tracked eight reference points (four cage corners and four cage tops) in the cage for normalization purposes. The same model was also trained on reference frames acquired using the BIO-T2CT system. The resulting model generalized well to both setups, ensuring comparability of the outputs.

The DeepLabCut predictions were then further curated by replacing bad predictions (<95% likelihood) with missing values, and removing recording sessions where >25% were missing (usually due to the animal escaping the inner cage). Furthermore, we removed sessions in which the animals showed excessive climbing behaviour (>95% of time between upper and lower cage corners). For the remaining sessions, the missing values were linearly interpolated and the centroid of the ears, body centre, and tail-base was used as the position of the animals. The X- and Y-positions of the centroid were then scaled to the tracked cage corner points to correct for minor movements of the camera or the setup. The resulting X- and Y-position time courses were then downsampled to 1 Hz. Only the first 30 minutes of the recording were kept. Shorter sessions were removed.

## Primary sensory neuron culture

Adult primary DRG cultures were prepared from 6- to 15-week-old animals as described previously (*Hanack et al., 2015*). Briefly, the animals were culled via isoflurane overdose, and their spinal columns excised and separated from muscle tissue. The spinal column was then cut lengthwise and the DRGs collected, freed from nerve branches, halved, and treated with a collagenase solution (1.25 mg/mL in Ringer's solution) for 1 hour at 37°C with gentle inversion every 15 minutes. This was followed by a 15-minute trypsin digestion (2.5 mg/mL) at 37°C, repeated trituration and suspension in complete culturing medium (DMEM/F12 w/o Glutamin, 10% heat-inactivated FCS, 2 mM L-glutamine (GlutaMAX), 1× anti-biotic/mitotic), and centrifugation at 900 rpm for 10 minutes over a BSA solution (150 mg/mL) to pellet the cells. The supernatant was discarded, the pellet was resuspended in culturing media, and spotted onto PDL- and Laminin-coated glass coverslips (5 mm). The cells were then left to settle onto the coverslip in the incubator for 1 hour at 37°C and then covered with culturing media. Cultures were either used the following day (overnight) or kept for 3 days, with a medium change after the first day.

## Calcium imaging recordings

For calcium imaging, cells cultured on coverslips were incubated for either 1 or 3 days. Before imaging, cells underwent a washing process with Ringer's solution (140 mM NaCl, 5 mM KCl, 2 mM $MgCl_2$, 2 mM $CaCl_2$, 10 mM glucose, and 10 mM HEPES, adjusted to pH 7.4), followed by loading with the calcium-sensitive dye Cal520-AM (10 µM) and Pluronic acid F-127 (0.05%) in Ringer's solution. The cells were incubated for 1 hours at 37°C, then the dye solution was replaced with Ringer's solution for a further 30 minutes at room temperature, minimizing light exposure.

The perfusion system, a ValveBank II (Automate) to control multiple inflows, and an air200 aquarium pump (Eheim) for outflow, was set to a maximum flow rate of 3 mL/min, facilitating laminar flow in the imaging chamber (RC-22, Warner Instruments). The coverslip was placed near the outlet to minimize movement artefacts. An IT-18 thermocouple (physitemp) was placed close to the coverslip to record the chamber temperature. Imaging settings varied based on the camera used: for CoolSnapHQ2 (Photometrics), exposure was set at 5 ms with 3× gain and 3 × 3 binning, whereas for Zyla 4.2 (Andor Technology), it was 80 ms with 2 × 2 binning. We used a Lambda DG-4 as a light source, maintained at 30% intensity to reduce bleaching.

Images were captured using MetaFluor software at 4 Hz or 10 Hz frequencies. Standard experiments involved a 1-minute baseline, 25-second stimuli, followed by a 3–5-minute recovery period with room temperature Ringer's solution (*Figure 2*, *Figure 2—figure supplement 2*). To identify TRPV1-positive neurons, we used the agonist Capsaicin (1 µM, *Caterina et al., 1997*). A Ringer's solution with a high potassium concentration (100 mM KCl) was used as a fina lstimulus to identify neurons. Solutions were heated via glass coils connected to a heated water bath. Each FOV was imaged for a maximum of 60 min, and the usage of coverslips was limited to 2 hours post-loading with the dye.

## Calcium imaging preprocessing and analysis

Calcium imaging data were motion-corrected and preprocessed using *Suite2p* (*Pachitariu et al., 2016*). Cell regions of interest (ROIs) were identified using the *Cellpose* package integrated into *Suite2p* (*Stringer et al., 2021*). The mean fluorescence of cells and surrounding neuropil was calculated by *Suite2p*, and neuropil contamination was corrected by subtracting 70% of the background neuropil traces from each cells fluorescence trace. The corrected data was then imported into a custom R-package, *neuroimgr*, for further analysis in R (https://github.com/hummuscience/neuroimgr, copy archived at *Abd El Hay, 2025*).

Normalization was performed using the $\Delta F/F_0$ method, where baseline fluorescence ($F_0$) is calculated as the mean fluorescence of the baseline, and $\Delta F$ is the change in fluorescence over time. For heatmaps, $F_0$ was estimated using the first 10 seconds of the experiments. For individual stimuli, the mean of the first 10 frames was used as $F_0$. Heatmaps were generated using the *ComplexHeatmap* R package. Cells were sorted by the earliest time point where they cross 10% of their cumulative $\Delta F/F_0$ in a given FOV and clustered using the Ward D2 algorithm. $\Delta F/F_0$ values smaller than the 0.1 and larger than the 99.9 percentile were clipped.

Due to the temperature sensitivity and loading variability of the calcium dye, a threshold-based approach failed to reliably identify responding cells across experimental days and FOVs (*Oliver et al., 2000*). Therefore, we used time-series classification to identify temperature-responsive cells. For this, calcium traces for each cell and stimulus were normalized, downsampled to 4 Hz, and a sample of 1000 traces across stimuli was manually labelled to create a training dataset. Examples of responsive and non-responsive cells as well as the average of each label are shown in *Figure 2—figure supplement 1A and B*. This ground-truth dataset was used to evaluate multiple time-series classification algorithms, with MINIROCKET (as implemented in the sktime Python package, *Dempster et al., 2021*) yielding the best results (classification results shown in *Figure 2—figure supplement 1B*). The trained classifier was then applied to the remaining cells to identify temperature responsive cells. A similar approach was used to identify capsaicin-responsive cells. Cells that did not respond to any of the applied stimuli were excluded from the analysis.

## Drift diffusion model

We employed a DDM to analyse the behaviour of mice in thermal chamber experiments. The DDM was preferred over simpler models like the Markov switching model as the latter did not provide satisfactory fits to our data (data not shown). Each parameter in the model can either be fit as a predictor (dependent on genotype, temperature combination, and temperature), fit to the entire data (floating), or fixed to a certain value. A typical DDM as described in *Ratcliff, 1978* can drift to the upper or lower bound (representing two choices), but our experimental design only offers one choice (to leave the chamber). To reduce the probability of reaching the lower bound and thereby improve the fit, we fixed the starting point/bias ($z$) to 0.9. This ensures that the evidence accumulation starts at a point that is much closer to the upper bound ($a$) than to the lower bound ($-a$). To choose the best combination of parameters that fits the data, we fit all the data to all remaining combinations of $v$, $sv$, and $a$

(*Figure 5—figure supplement 1A and B*) and compared them via the expected log pointwise predictive density (ELPD) by Pareto smoothed importance sampling leave-one-out cross-validation (LOO) (*Vehtari et al., 2017*). Unreliable models as per *Vehtari et al., 2017* were discarded. The chosen model was constructed to account for variations in both noise (*sv*) and drift rate (*v*) for each genotype, temperature comparison, and chamber/temperature (*Figure 5—figure supplement 1A and B*, and *Equation 1*). To accommodate individual differences among animals, we introduced a random effect for each animal in the model. This approach enabled us to capture the unique behavioural patterns of each subject while assessing the general trends across the population.

$$v \mid sv \sim \text{temperature combination} \times \text{chamber} \times \text{genotype} + (1 \mid \text{animal}) \qquad (1)$$

To fit the model, we applied a hierarchical Markov chain Monte Carlo (MCMC) sampling approach as implemented in the *HSSM* python package. For the implementation of the hierarchical MCMC, we utilized the No-U-Turn Sampler (NUTS) as implemented in *NumPyro*, a robust algorithm for efficiently sampling from high-dimensional probability distributions. The tuning phase for all fit models involved 2000 samples, ensuring adequate exploration of the parameter space and helping to achieve convergence. The final model was run with four chains, each drawing 2000 samples.

## Statistical methods

Statistical analyses were conducted using R software. For time-course experiments involving repeated measures, two-way ANOVA with repeated measures as implemented in the *afex* package was conducted. Mauchly's test was applied to assess the assumption of sphericity, and corrections for violations were made using the Geisser–Greenhouse correction. For comparisons with imbalanced observations or missing data, we fit a linear mixed effect model to the data using the *lmer* package. In cases of significant outcomes, post hoc comparisons were performed using estimated marginal means (EMMs) with pairwise contrasts comparing treatments to control groups (facilitated by the *emmeans* package), with false discovery rate (FDR) for multiple comparison corrections. For non-parametric data, we applied a Wilcoxon rank-sum test as implemented in the *rstatix* package, coupled with FDR for multiple comparison correction. To assess differences in crossing behaviour, a Cox proportional hazard model was used, as implemented in the *survival* package. For visit length comparisons, a mixed-effects model was fit using the *lme4* package, allowing random effects for animal subjects where appropriate and correcting for the effect of time. Multiple comparisons for the mixed-effects and Cox models were accounted for using the FDR approach within the *emmeans* and *multcomp* packages. Only statistically significant results ($p < 0.05$) are shown.

## Resource availability

### Lead contact

Requests for resources and reagents should be directed to and will be fulfilled by the lead contact Jan Siemens (jan.siemens@pharma.uni-heidelberg.de).

## Materials availability

This study did not generate any unique reagents.

## Acknowledgements

We thank Christina Steinmeier-Stannek, Annika von Seggern, Daniela Pimonov, Lisa Vierbaum, Amandine Cavaroc, and Lisa Weiler for technical support; members of the Siemens lab, particularly Hagen Wende, Katrin Schrenk-Siemens, and Jrg Pohle, for inspiring discussions and critical input. Additionally, we thank Juan Boffi, Katharine Shapcott, Natalie Schaworonkow, Marieke Schlvinck, and Martha Nari-Havenith for their support, discussions, scientific input, and valuable criticism of the work. We gratefully acknowledge the data storage service SDS@hd supported by the Ministry of Science, Research and the Arts Baden-Wrttemberg (MWK) and the German Research Foundation (DFG) through grant INST 35/1314-1 FUGG. ATB acknowledges support from the Margarita Salas Fellowship and the Joachim Herz Stiftung. This work was supported by the European Research Council ERC-CoG-772395 and the German Research Foundation SFB/TRR 152 and SFB1158 (to JES).

## Additional information

### Funding

| Funder | Grant reference number | Author |
|---|---|---|
| European Research Council | 10.3030/772395 | Jan Siemens |
| Deutsche Forschungsgemeinschaft | SFB 1158 | Jan Siemens |
| Deutsche Forschungsgemeinschaft | SFB/TRR 152 | Jan Siemens |
| Joachim Herz Stiftung | | Alejandro Tlaie Boria |
| Margarita Salas Foundation | Margarita Salas Fellowship | Alejandro Tlaie Boria |

The funders had no role in study design, data collection and interpretation, or the decision to submit the work for publication.

### Author contributions

Muad Y Abd El Hay, Conceptualization, Resources, Data curation, Software, Formal analysis, Validation, Investigation, Visualization, Methodology, Writing – original draft, Project administration, Writing – review and editing; Gretel B Kamm, Data curation, Formal analysis, Supervision, Funding acquisition, Investigation, Methodology, Writing – review and editing; Alejandro Tlaie Boria, Formal analysis, Investigation, Visualization, Writing – review and editing; Jan Siemens, Conceptualization, Resources, Supervision, Funding acquisition, Writing – original draft, Project administration, Writing – review and editing

### Author ORCIDs

Muad Y Abd El Hay ⓘ https://orcid.org/0000-0002-5082-1216
Jan Siemens ⓘ https://orcid.org/0000-0001-9051-9217

### Ethics

All animal care and experimental procedures were approved by the local council (Regierungspräsidium Karlsruhe, Germany) under protocol numbers G-201/16 and T05-19.

Reviewer #3 (Public review): https://doi.org/10.7554/eLife.95618.4.sa1
Author response https://doi.org/10.7554/eLife.95618.4.sa2

## Additional files

### Supplementary files

MDAR checklist

### Data availability

The datasets and code supporting the study have been deposited in a public repository (https://heidata.uni-heidelberg.de/) and can be accessed using the following DOI: https://doi.org/10.11588/DATA/8VKA7I. Any additional information required to reanalyse the data reported in this article is available from the corresponding author, Muad Abd El Hay, upon request.

The following dataset was generated:

| Author(s) | Year | Dataset title | Dataset URL | Database and Identifier |
|---|---|---|---|---|
| Abd el Hay MY, Kamm GB, Tlaie A, Siemens J | 2025 | Diverging roles of TRPV1 and TRPM2 in warm-temperature detection | https://doi.org/10.11588/DATA/8VKA7I | heiDATA, 10.11588/DATA/8VKA7I |

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
