## [Editor Report · eLife Assessment]

In this article, Abd El Hay and colleagues use an innovative behavioural assay and analysis method, together with standard calcium imaging experiments on cultured dorsal root ganglion (DRG) neurons, to evaluate the consequences of global knockout of TRPV1 and TRPM2, and overexpression of TRPV1, on warmth detection. **Compelling** evidence is provided for a role of TRPM2 channels in warmth avoidance behaviour, but it remains unclear whether this involves channel activity in the periphery or in the brain. In contrast, TRPV1 is clearly implicated at the cellular level in warmth detection. These findings are **important** because there is substantial ongoing discussion regarding the contribution of TRP channels to different aspects of thermo-sensation.

---

## [Referee Report · Reviewer #3 (Public review)]

A central question in the thermal system is which thermally responsive ion channels are responsible for warm evoked behaviors and DRG afferent neuron responses to warming. Recent work has shown evidence for TRPV1, TRPM2 and TRPM8. Here Abd El Hay and colleagues investigate the role of TRPM2 and TRPV1 in a novel warm preference behavior and in the thermal responses of cultured DRG neurons.

They develop a new thermal preference task, where both the floor and air temperature are controlled, which shows differences to the classic two-plate preference task. This is a central strength of the paper, as it will allow a new method to investigate how animals integrating floor and air temperature. They go on to use knockout mice and confirm a clear role for TRPM2 in warm preference behavior.

Using a new approach for culturing DRG neurons they investigate the involvement of both channels in warm responsiveness and dynamics. In apparent contrast to the role of TRPM2 on thermal behavior, it does not have a major effect on the responses of cultured DRG neurons to warm stimuli. Eliminating TRPV1 however has a stronger impact on DRG responses, particularly at low stimulus amplitudes. It will be important to discover how TRPM2 influences warm driven behaviors, if it is not via changes in afferent response properties.

Thanks to the authors for addressing my remaining questions in this updated version of the manuscript.

This is an interesting study with novel approaches that generates new information on the differing roles of TRPV1 and TRPM2 on thermal behavior.

---

## [Author Response]

The following is the authors’ response to the previous reviews.

**Public Reviews:**

**Reviewer # 1 (Public Review):**
Summary:The authors use an innovative behavior assay (chamber preference test) and standard calcium imaging experiments on cultured dorsal root ganglion (DRG) neurons to evaluate the consequences of global knockout of TRPV1 and TRPM2, and overexpression of TRPV1, on warmth detection. They find a profound effect of TRPM2 elimination in the behavioral assay, whereas the elimination of TRPV1 has the largest effect on the neuronal responses. These findings are very important, as there is substantial ongoing discussion in the field regarding the contribution of TRP channels to different aspects of thermosensation.Strengths:The chamber preference test is an important innovation compared to the standard two-plate test, as it depends on thermal information sampled from the entire skin, as opposed to only the plantar side of the paws. With this assay, and the detailed analysis, the authors provide strong supporting evidence for a role of TRPM2 in warmth avoidance. The conceptual framework using the Drift Diffusion Model provides a first glimpse of how this decision of a mouse to change between temperatures can be interpreted and may form the basis for further analysis of thermosensory behavior.Weaknesses:The authors juxtapose these behavioral data with calcium imaging data using isolated DRG neurons. As the authors acknowledge, it remains unclear whether the clear behavioral effect seen in the TRPM2 knockout animals is directly related to TRPM2 functioning as a warmth sensor in sensory neurons. The effects of the TRPM2 KO on the proportion of warmth sensing neurons are very subtle, and TRPM2 may also play a role in the behavioral assay through its expression in thermoregulatory processes in the brain. Future behavioral experiments on sensory-neuron specific TRPM2 knockout animals will be required to clarify this important point.
**Reviewer # 1 (Recommendations for the authors):**
(1) I have no further suggestions for the authors, and congratulate them with their excellent study.For the authors information, ref. 42 does contain behavioral data from both male (Fig. 4 and Extended Figure 7) and female (Extended Figure 8) mice.

We thank the referee for pointing out that both males and female mice were tested in the Vandewauw et al. 2018 study. We deliberated whether to include this at the appropriate section of our manuscript (“Limitations of the Study”). But since Vandewauw et al. assessed noxious heat temperatures and we here assess innocuous warmth temperature, we felt that this reference would not add to the clarification whether there are sex differences in Trp channelbased warmth temperature sensing. In particular, we did not want to “use” the argument and to suggest that there are no sex temperature differences in the warmth range just because Vandewauw et al. did not observe major sex differences in the noxious temperature range.

**Reviewer #3 (Public Review):**
Summary and strengths:In the manuscript, Abd El Hay et al investigate the role of thermally sensitive ion channels TRPM2 and TRPV1 in warm preference and their dynamic response features to thermal stimulation. They develop a novel thermal preference task, where both the floor and air temperature are controlled, and conclude that mice likely integrate floor with air temperature to form a thermal preference. They go on to use knockout mice and show that TRPM2-/- mice play a role in the avoidance of warmer temperatures. Using a new approach for culturing DRG neurons they show the involvement of both channels in warm responsiveness and dynamics. This is an interesting study with novel methods that generate important new information on the different roles of TRPV1 and TRPM2 on thermal behavior.Comments on revisions:Thanks to the authors for addressing all the points raised. They now include more details about the classifier, better place their work in context of the literature, corrected the FOVs, and explained the model a bit further. The new analysis in Figure 2 has thrown up some surprising results about cellular responses that seem to reduce the connection between the cellular and behavioral data and there are a few things to address because of this:(1) TRPM2 deficient responses: The differences in the proportion of TRPM2 deficient responders compared to WT are only observed at one amplitude (39C), and even at this amplitude the effect is subtle. Most surprisingly, TRPM2 deficient cells have an enhanced response to warm compared to WT mice to 33C, but the same response amplitude as WT at 36C and 39C. The authors discuss why this disconnect might be the case, but together with the lack of differences between WT and TRPM2 deficient mice in Fig 3, the data seem in good agreement with ref 7 that there is little effect of TRPM2 on DRG responses to warm in contrast to a larger effect of TRPV1. This doesn't take away from the fact there is a behavioral phenotype in the TRPM2 deficient mice, but the impact of TRPM2 on DRG cellular warm responses is weak and the authors should tone down or remove statements about the strength of TRPM2's impact throughout the manuscript, for example:"Trpv1 and Trpm2 knockouts have decreased proportions of warm-sensitive neurons.""this is the first cellular evidence for the involvement of TRPM2 on the response of DRG sensory neurons to warm-temperature stimuli""we demonstrate that TRPV1 and TRPM2 channels contribute differently to temperature detection, supported by behavioural and cellular data""TRPV1 and TRPM2 affect the abundance of warm-sensitive neurons, with TRPV1 mediating the rapid, dynamic response to warmth and TRPM2 affecting the population response of warm-sensitive neurons.""Lack of TRPV1 or TRPM2 led to a significant reduction in the proportion of warm-sensitive neurons, compared to wildtype cultures".

We agree with the referee that the somewhat surprising result of the subtle phenotype in Trpm2 knock-out DRG culture experiments, that became detectable in the course of the new analysis, was overemphasized in the previous version of the manuscript. Per suggestion, we have toned down or removed the statements in the revised manuscript (for the referee to find those changes easily, they are indicated in “track-changes mode” in the submitted document).

(2) The new analysis also shows that the removal of TRPV1 leads to cellular responses with smaller responses at low stimulus levels but larger responses with longer latencies at higher stimulus levels. Authors should discuss this further and how it fits with the behavioral data.

Because these changes shown in Fig. 2E are also subtle (similar to the cellular Trpm2 phenotype discussed above), and because both the “% Responders” (Fig 2.D) and The AUC analysis (Fig. 2F) show a reduction in Trpv1 knock out cultures ––both, at lower and at higher stimulus levels–– we did not want to overstate this difference too much and therefore did not further discuss this aspect in the context of the behavioral differences observed in the Trpv1 knock-out animals.

(3) Analysis clarification: authors state that TRPM2 deficient warm-sensitive neurons show "Their response to the second and third stimulus, however, are similar to wildtype warm-sensitive neurons, suggesting that tuning of the response magnitude to different warmth stimuli is degraded in Trpm2-/- animals." but is there a graded response in WT mice? It looks like there is in terms of the %responders but not in terms of response amplitude or AUC. Authors could show stats on the figure showing differences in response amplitude/AUC/responders% to different stimulus amplitudes within the WT group.

We have added the statistics in the main text, you find them on page 7 (also in “track changes mode”).

(4) New discussion point: sex differences are "similar to what has been shown for an operant-based thermal choice assay (11,56)", but in their rebuttal, they mention that ref 11 did not report sex differences. 56 does. Check this.

Thank you for pointing out this mishap. We have now corrected this in the “Limitations of the study” section of the discussion and have removed the Paricio-Montesions et al study from that section and slightly revised the text (see “track-changes” on page 16).

(5) The authors added in new text about the drift diffusion model in the results, however it's still not completely clear whether the "noise" is due to a perceptual deficit or some other underlying cause. Perhaps authors could discuss this further in the discussion.

We have now included more discussion concerning this (page 14):

“However, the increased noise in the drift-di3usion model points to a less reliable temperature detection mechanism. Although noise in drift di3usion models can encompass various sources of variability—ranging from peripheral sensory processing to central mechanisms like attention or motor initiation—the most parsimonious interpretation in our study aligns with a perceptual deficit, given the altered temperatureresponsive neuronal populations we observed. This implies that, despite the substantial loss of warm-sensitive neurons, the remaining neuronal population provides su3icient information for the detection of warmer temperatures, albeit with reduced precision”

Within the limits of the data that is available, we hope the referee agrees with us that we have now adequately discussed this aspect; we feel that any further discussion would be too speculative.